# Distributionally Robust Models
# with Parametric Likelihood Ratios

**Paul Michel**
Centre Sciences des Données
École normale supérieure PSL
Paris, 75005, France
`pmichel31415@gmail.com`

**Tatsunori Hashimoto**
Computer Science Department
Stanford University
Stanford, CA 94305, USA
`thashim@stanford.edu`

**Graham Neubig**
School of Computer Science
Carnegie Mellon University
Pittsburgh, PA 15213, USA
`gneubig@cs.cmu.edu`

## Abstract

As machine learning models are deployed ever more broadly, it becomes increasingly important that they are not only able to perform well on their training distribution, but also yield accurate predictions when confronted with distribution shift. The Distributionally Robust Optimization (DRO) framework proposes to address this issue by training models to minimize their expected risk under a collection of distributions, to imitate test-time shifts. This is most commonly achieved by instance-level re-weighting of the training objective to emulate the likelihood ratio with possible test distributions, which allows for estimating their empirical risk via importance sampling (assuming that they are subpopulations of the training distribution). However, re-weighting schemes in the literature are usually limited due to the difficulty of keeping the optimization problem tractable and the complexity of enforcing normalization constraints. In this paper, we show that three simple ideas – mini-batch level normalization, a KL penalty and simultaneous gradient updates – allow us to train models with DRO using a broader class of parametric likelihood ratios. In a series of experiments on both image and text classification benchmarks, we find that models trained with the resulting parametric adversaries are consistently more robust to subpopulation shifts when compared to other DRO approaches, and that the method performs reliably well with little hyper-parameter tuning.[1]

## 1 Introduction

It is well acknowledged that modern neural network based machine learning models tend to underperform when they are evaluated on data distributions that differ from the one they were trained on. For example, machine learning model performance has been observed to degrade under train-test mismatch in topics (Gururangan et al., 2020), demographics (Blodgett et al., 2016; Amodei et al., 2016; Hovy & Søgaard, 2015; Grother et al., 2019), geographic regions (Koh et al., 2020), and even data collection processes (Beery et al., 2018; Zech et al., 2018; Michel & Neubig, 2018). In particular, these models often perform poorly when evaluated on subpopulations, domains that are present but underrepresented in their training data (Sagawa et al., 2020), and they can latch on to spurious correlations (McCoy et al., 2019). This has problematic real-world consequences: when such models are deployed at large, this representation disparity can, for example, unfairly affect minority groups (Buolamwini & Gebru, 2018; Hashimoto et al., 2018).

This behaviour can largely be attributed to the empirical risk minimization (ERM) principle which underlies the majority of learning algorithms used in practice. In ERM, models are trained to minimize the average loss over a finite sample from a fixed training distribution (Vapnik, 1992), as a proxy for the expected loss on a random example drawn from the fixed, albeit unknown, data distribution. This favors models which perform well *on average* on a fixed training set, as opposed to models which would perform equally well on a variety of subpopulations that better reflects the diverse set of distributions that can be encountered at test time. On the other hand, Distributionally robust optimization (DRO) proposes an appealing alternative to ERM. In DRO, models are trained to

---

[1]Code to reproduce our experiments can be found at `https://github.com/pmichel31415/P-DRO`

minimize their worst case risk (or an empirical estimate thereof computed on a finite sample, via *e.g.* importance weighting) under a pre-determined family of distributions $\mathcal{Q}$, called the "uncertainty set" (or "ambiguity set"):

$$\mathcal{L}_{\text{DRO}}(\theta) = \max_{q \in \mathcal{Q}} \mathbb{E}_{(x,y) \sim q} \ell_\theta(x, y). \tag{1}$$

In the absence of explicit information about the subpopulations of interest (which would naturally define $\mathcal{Q}$), it is up to the practitioner to carefully define this uncertainty set. This has been the subject of much work in the literature (see Rahimian & Mehrotra (2019) for a survey). Recently, Michel et al. (2021) proposed P-DRO, a promising approach where the uncertainty set is defined by a parametric family of generative models, which allows for more flexibility in defining the uncertainty set. P-DRO shows significant improvement over comparable baselines, but it suffers from several drawbacks. First, it presupposes the availability of generative models capable of outputting exact densities, which limits its field of application to modalities where such models are readily available (such as language models in NLP). Second, it is challenging to use in practice due to its reliance on a number of hyper-parameters and approximations to the objective function.

In this paper, we propose a new approach for DRO, called RP-DRO, based on a key modification of the P-DRO algorithm: instead of modeling the worst-case distributions directly, we parametrize the likelihood ratio between the training distribution and the worst-case distribution. This removes the dependency on an unwieldy generative model, making the method useful for more applications. While likelihood ratio formulations of DRO have been tried in prior work (Sagawa et al., 2020), we show that they are particularly effective for parametric, neural network based adversaries. Our approach relies on three simple ideas: a mini-batch level normalization strategy to enforce likelihood ratio constraints, a penalty-form of the KL divergence uncertainty set, and the use of simultaneous gradient updates for training. RP-DRO consistently achieves equal or better robust subpopulation accuracy compared to P-DRO and other baselines on a variety of standard benchmarks in image and text classification. In addition, we find it is both faster than P-DRO and depends on fewer hyper-parameters. Additional ablation experiments demonstrate that both our mini-batch normalization strategy and simultaneous gradient updates are necessary for high performance. Finally, we perform experimental analyses to shed light on the advantages brought by parametric adversaries compared to their nonparametric counterparts.

## 2 BACKGROUND

In the following, we consider a model parametrized by $\theta \in \mathbb{R}^{d_{\text{model}}}$. Our goal is to find a model which minimizes the loss function $\ell_\theta(x, y)$ on pairs of inputs and outputs $(x, y) \in \mathcal{X} \times \mathcal{Y}$. For instance, $x$ might represent images and $y$ a categorical label. Parameters $\theta$ are estimated on a training dataset $\mathcal{D}_{\text{train}} = \{(x_i, y_i)\}_{i=1...N_{\text{train}}}$ which we assume to be drawn from a training distribution $p$.

The DRO optimization problem with uncertainty set $\mathcal{Q}$ is

$$\min_\theta \max_{q \in \mathcal{Q}} \mathbb{E}_q \ell_\theta(x, y). \tag{2}$$

Note that the DRO loss in Eq. 1 is the inner maximum of the DRO problem, and it provides an upper bound on the expected loss of the model under any distribution in the uncertainty set $\mathcal{Q}$. This motivates the use the minimizer of the min-max game in Eq. 2 as a robust model. We refer to the solution of the inner maximum as the "adversary" from now on

However this objective is only useful insofar that (1) $\mathcal{Q}$ covers test distributions of interest (corresponding to different domains, demographics, etc.) and (2) $\mathcal{Q}$ is not overly pessimistic. To fulfil this second condition, there should exist some model $\theta^*$ that achieves low loss simultaneously on the test distribution as well as $\mathcal{Q}$. This often requires that $\mathcal{Q}$ only contain distributions that are covariate shifts of the test distribution, *i.e.* that are such that the conditional distribution $q(y \mid x)$ coincides with that of training distribution $p(y \mid x)$.

### 2.1 NONPARAMETRIC DRO

There is substantial existing work on *nonparametric* formulations of DRO, where $\mathcal{Q}$ is expressed as a divergence ball centered at the training distribution. This includes $f$-divergences (Ben-Tal et al., 2013; Hu & Hong, 2013; Faury et al., 2020), Wasserstein/IPM (Sinha et al., 2018; Husain, 2020), moment constraints (Delage & Ye, 2010; Nguyen et al., 2020), and CVaR (Fan et al., 2017; Curi et al., 2020;

Levy et al., 2020) based uncertainty sets. These nonparametric approaches are appealing as they require very little domain-specific knowledge, have well-understood theory (Duchi & Namkoong, 2018), and optimization procedures (*e.g.* Hu & Hong (2013) for KL constraints and Levy et al. (2020) for $\chi^2$ and CVaR constraints).

Unfortunately, nonparametric DRO algorithms suffer from being overly pessimistic. Their uncertainty sets tend to include distributions that are exceedingly difficult to learn, or not representative of real-world distribution shifts. Furthermore, they often cannot enforce even basic constraints such as covariate shift structures (Duchi et al., 2020; Hu et al., 2018). Group-structured DRO uncertainty sets (Sagawa et al., 2020) overcome some of these challenges, but require significant domain expertise to pre-specify target subpopulations that a model should be robust to.

## 2.2 PARAMETRIC DRO

Parametric DRO (Michel et al., 2021) is a method for DRO in which the uncertainty set $\mathcal{Q}$ is defined as a family of parametric generative models, which avoids the extreme pessimism of nonparametric DRO without the explicit specification of subpopulations. Specifically, given a generative model $q_\psi$ parameterized by $\psi \in \mathbb{R}^{d_{\text{adv}}}$, the KL-constrained parametric DRO objective can be written as follows:

$$\min_\theta \max_{\substack{\psi \\ \text{KL}(q_\psi \| p) \leq \kappa}} \mathbb{E}_{(x,y) \sim q_\psi} \ell(x, y, \theta). \tag{3}$$

As demonstrated by Michel et al. (2021), P-DRO yields significant improvements over its nonparametric counterpart. However, the difficulty of optimizing Eq. 3 directly results in a number of approximations and additional hyper-parameters that are hard to tune.

In addition, a central drawback of P-DRO is that it necessitates training an auxiliary generative model of the data. This can be difficult for several reasons. First, this limits the applicability of the method to domains with generative models that allow for exact probability computations. Moreover, even when such generative models are available, they are often more computationally demanding than their discriminative counterparts. In language models for instance, probabilities for sequences of text are obtained by iteratively producing conditional probabilities over all tokens in the vocabulary. This additional step results in considerable computational overhead compared to discriminative models.

## 3 PARAMETRIC LIKELIHOOD RATIO

### 3.1 DRO AS A LIKELIHOOD RATIO OPTIMIZATION PROBLEM

In the situation that all distributions in $\mathcal{Q}$ are absolutely continuous with respect to $p$ (i.e. for all measurable subset $A \subset \mathcal{X} \times \mathcal{Y}$, all $q \in \mathcal{Q}$, $q(A) > 0$ only if $p(A) > 0$) the inner maximum in Eq. 2 can be rewritten purely as a function of the likelihood ratio $\frac{q}{p}$

$$\mathbb{E}_{(x,y) \sim q} \ell_\theta(x, y) = \mathbb{E}_{(x,y) \sim p} \frac{q(x, y)}{p(x, y)} \ell_\theta(x, y). \tag{4}$$

Such absolute continuity assumptions are standard in $f$-divergence and group DRO methods, which both rely upon re-weighting the training distributions. In fact, the KL divergence constraint in P-DRO presupposes absolute continuity.

This suggests that the inner maximum can be re-written as an optimization problem on functions $r : \mathcal{X} \times \mathcal{Y} \longrightarrow \mathbb{R}_+$ within the uncertainty set $\mathcal{R} \in \{r \mid pr \in \mathcal{Q}\}$

$$\min_\theta \max_{r \in \mathcal{R}} \mathbb{E}_{(x,y) \sim p} r(x, y) \ell_\theta(x, y). \tag{5}$$

This reparametrization of the problem will allow us to replace a parametric family of generative models with a parametric family over probability ratios.

### 3.2 RATIO-BASED P-DRO

The likelihood ratio formulation described above is appealing for P-DRO because it enables the use of discriminative style neural architectures for parametrizing the ratio $r$, which opens up many more

options for defining the parametric uncertainty set. Specifically, we can set the adversary to be any parametric function $r_\psi : \mathcal{X} \times Y \longrightarrow \mathbb{R}^+$ verifying $\mathbb{E}_{x,y \sim p}\, r_\psi(x, y) = 1$. The key insight that we use to realize our proposed method is that we do not need to restrict the choice of adversaries to those that implicitly satisfy this normalization condition (*i.e.* generative models). Instead, we can pick any adversary and treat normalization as an additional constraint (the "normalization constraint").

Note that in this case, the KL constraint takes the simple form $\mathrm{KL}(pr_\psi \| p) = \mathbb{E}_{pr_\psi} \log \frac{pr_\psi}{p} = \mathbb{E}_p\, r_\psi \log r_\psi$. The final min-max problem, which we dub ratio-based P-DRO (RP-DRO), is:

$$\min_\theta \max_{\substack{\psi \\ \mathbb{E}_p\, r_\psi \log r_\psi \leq \kappa \\ \mathbb{E}_p r_\psi = 1}} \underbrace{\mathbb{E}_{(x,y) \sim p} r_\psi(x,y) \ell_\theta(x,y)}_{\mathcal{L}_{\text{RP-DRO}}}. \tag{6}$$

As in P-DRO, we can look for equilibria of this differentiable min-max game by performing simultaneous gradient updates (Singh et al., 2000) to $\theta$ and $\psi$ in directions $-\nabla_\theta \mathcal{L}_{\text{RP-DRO}}$ and $+\nabla_\psi \mathcal{L}_{\text{RP-DRO}}$ respectively. Although finding global equilibria is not guaranteed in high dimensional non-convex settings (Balduzzi et al., 2018), empirical evidence suggests that models trained in this manner still reach useful solutions (Michel et al., 2021).

In experiments, we adopt an exponential parametrization $r_\psi(x, y) \propto e^{f_\psi(x,y)}$ where $f_\psi$ is the output of any parametric model with values in $\mathbb{R}$. Similarly to P-DRO, we do not explicitly enforce the KL constraint (due to the difficulty of projecting onto the KL ball), and instead we relax it in the form of a term $\tau\, \mathbb{E}_p\, r_\psi \log r_\psi$ added to the loss function. The regularization strength $\tau$ is treated as a hyper-parameter.

### 3.3 ENFORCING THE NORMALIZATION CONSTRAINT

In addition to the KL constraint, RP-DRO necessitates that $r_\psi$ satisfies a normalization constraint $\mathbb{E}_p\, r_\psi = 1$ to ensure that $pr_\psi$ is a proper probability distribution over $\mathcal{D}_{\text{train}}$. If that were not the case, the adversary $r_\psi$ could artificially increase the weighted expectation $\mathbb{E}_p\, r_\psi \ell_\theta$ by assigning a total weight greater than 1 to the entire dataset.

Existing methods for ratio based DRO such as Sagawa et al. (2020) achieve this by either projecting $r_\psi$ onto the set $\{r \mid \mathbb{E}_p\, r = 1\}$ after each gradient update on $\psi$, or by directly parametrizing the ratio as $r_\psi(x, y) = e^{f_\psi(x,y)} / \mathbb{E}_p\, e^{f_\psi}$. Unfortunately, these solutions are computationally infeasible in practical scenarios with large datasets. Indeed, they necessitate computing the entire expectation over $p$, which can be costly when each $f_\psi(x, y)$ is the output of a neural model.

We propose two simple, yet effective solutions for addressing this issue in the context of mini-batch training where we can only compute $f_\psi$ for small number of samples $(x_1, y_1), \ldots, (x_n, y_n)$ at each step of training.

**Self-normalization** is inspired by the idea of "self-normalization" developed for globally normalized structured prediction models (Andreas et al., 2015; Goyal et al., 2019). It consists in adding a relaxation of the normalization constraint to the objective. Specifically, following Goyal et al. (2019) we add a squared penalty on the log normalizer at the mini-batch level. Ignoring the KL penalty, this regularized objective takes the following empirical form:

$$\hat{\mathcal{L}}_{\text{self-norm}}(\theta, \psi) = \frac{1}{n} \sum_{i=1}^n r_\psi(x_i, y_i) \ell_\theta(x_i, y_i) - \beta \log \left( \frac{1}{n} \sum_{i=1}^n r_\psi(x_i, y_i) \right)^2. \tag{7}$$

The hyper-parameter $\beta$ controls the regularization strength. Intuitively, this penalizes adversaries that assign too much (or too little) total weight to the mini-batch. However, the choice of an optimal $\beta$ adds an additional degree of freedom to RP-DRO, which suggests our second option as a simpler alternative.

**Batch-level normalization** consists of using the normalized ratio at the mini-batch level by setting

$$\tilde{r}_\psi(x_i, y_i) = \frac{e^{f_\psi(x_i, y_i)}}{\sum_{j=1}^n e^{f_\psi(x_j, y_j)}} \tag{8}$$

for each sample $(x_i, y_i)$ in the mini-batch. An obvious downside of this approach is that the weight of each sample now depends on the mini-batch it was sampled from. This can be problematic for

small batch sizes: as an extreme example, for a batch size of 1, this normalization scheme assigns the same weight of 1 to every sample, making the objective equivalent to ERM.

However, mini-batch approximations have proven effective for other forms of DRO (Hu et al., 2018; Levy et al., 2020) and there is some evidence that they can yield accurate estimates for higher batch sizes (Cortes et al., 2010). In practice we find that this approach yields good results for commonly used batch sizes, is generally more stable than the self-normalization penalty, and does not introduce the additional hyper-parameter $\beta$. In most of our experiments, we adopt this approach unless specified otherwise. In that case, the empirical RP-DRO objective on a mini-batch becomes

$$\hat{\mathcal{L}}_{\text{batch-level norm}}(\theta, \psi) = \underbrace{\sum_{i=1}^{n} \tilde{r}_\psi(x_i, y_i) \ell_\theta(x_i, y_i)}_{\text{expected loss under } pr_\psi} - \tau \underbrace{\sum_{i=1}^{n} \tilde{r}_\psi(x_i, y_i) \log \tilde{r}_\psi(x_i, y_i)}_{\text{KL penalty}}. \qquad (9)$$

The KL term serves to penalize ratios that deviate too much from 1 (Note that the penalty is subtracted because we are maximizing with respect to $\psi$). The only hyper-parameter that needs to be tuned is the KL regularization strength $\tau$.

## 4 EXPERIMENTS

### 4.1 DATASETS

We perform experiments on four datasets: two text classification datasets used in Michel et al. (2021), BiasedSST and FDCL18, and two image classification datasets from Sagawa et al. (2020), Waterbirds and CelebA. Specific details for each dataset follow these previous works, as described below:

**BiasedSST** is based on the SST-2 sentiment classification dataset (Radford et al., 2018), but modified to introduce spurious correlation between a distractor token ("So,") and positive labels in around 95% of the dataset. In this setting models trained with ERM can very easily learn this spurious correlation, which hurts performance on the small subpopulation that doesn not suffer from this bias.

**FDCL18** A toxicity detection dataset of tweets labeled as *hateful* (5%), *abusive* (27%), *normal* (54%) and *spam* (14%). The group-DRO problem is formulated by partitioning the evaluation data along labels as dialectal annotation obtained automatically with an off-the shelf classifier (Blodgett et al., 2016; Sap et al., 2019). In particular these dialects align closely with self-reported race, and Sap et al. (2019) found that machine learning models trained on such toxicity detection datasets tend to exhibit bias towards certain labels depending on the demographics of the tweets' authors, particularly with minorities. In order to report more reliable accuracy numbers, all groups containing less than 100 samples are aggregated when computing test accuracies.

**Waterbirds** An image classification dataset where the task is to predict "land bird" or "water bird" with the confounding factor of the background; most water (resp. land) bird pictures have water (resp. land) on the background.

**CelebA** A popular face recognition dataset originally published by Liu et al. (2015). The group-DRO problem is formulated as a task of predicting the hair color ("Blond" or "Dark") across groups formed by the combination of the label and the (binary) gender of the subject. Due to the spurious correlation between blond hair/female and dark hair/male, models trained with ERM tend to achieve lower accuracies on underrepresented groups such as "blond-haired male" which totals only 0.85% of the training data.

### 4.2 EXPERIMENTAL SETTING

On BiasedSST and FDCL18 we follow Michel et al. (2021) and train a BiLSTM and BERT-base model Devlin et al. (2018) respectively. On the image classification datasets we train Resnet-50 architectures (He et al., 2016) pre-trained on ImageNet (Deng et al., 2009) as in Sagawa et al. (2020).

Since the adversary in RP-DRO can be any discriminative architecture, we opt for the natural solution of using a similar architecture for this model. For instance on BiasedSST, we take $f_\psi(x, y)$ as the

Table 1: Robust and average test accuracies on the Biased SST and FDCL18 datasets. Underlined numbers indicates statistically significant difference compared to ERM ($p$-value$< 0.05$). Bold numbers indicates the best number in each column (barring Oracle DRO).

| | Biased SST | | FDCL18 | |
| --- | --- | --- | --- | --- |
| | Robust | Average | Robust | Average |
| ERM | $2.15 \pm 0.97$ | $\mathbf{95.09} \pm 0.16$ | $19.57 \pm 7.00$ | $\mathbf{81.56} \pm 0.26$ |
| Topic DRO | $\underline{5.18} \pm 1.46$ | $95.00 \pm 0.10$ | $16.48 \pm 5.46$ | $\underline{80.49} \pm 0.49$ |
| NonParam-$\chi^2$ | $\underline{4.65} \pm 1.61$ | $95.10 \pm 0.22$ | $15.70 \pm 2.56$ | $81.55 \pm 0.23$ |
| NonParam-CVaR | $\underline{24.42} \pm 6.82$ | $\underline{89.03} \pm 3.73$ | $11.19 \pm 7.64$ | $\underline{69.19} \pm 8.28$ |
| NonParam-KL | $\underline{28.11} \pm 2.16$ | $\underline{92.45} \pm 1.55$ | $\underline{17.54} \pm 6.41$ | $\underline{81.20} \pm 0.11$ |
| P-DRO | $\underline{34.98} \pm 9.39$ | $\underline{84.21} \pm 2.11$ | $30.25 \pm 10.13$ | $79.91 \pm 1.41$ |
| RP-DRO | $\underline{\mathbf{50.70}} \pm 7.33$ | $\underline{86.60} \pm 2.96$ | $\underline{\mathbf{53.52}} \pm 1.66$ | $\underline{76.62} \pm 1.43$ |
| Oracle DRO | $\underline{67.71} \pm 3.03$ | $\underline{77.91} \pm 4.49$ | $\underline{55.23} \pm 3.97$ | $\underline{72.43} \pm 2.61$ |

Table 2: Robust and average test accuracies on the Waterbirds and CelebA datasets. Underlined numbers indicates statistically significant difference compared to ERM ($p$-value$< 0.05$). Bold numbers indicates the best number in each column (barring Oracle DRO).

| | Waterbirds | | CelebA | |
| --- | --- | --- | --- | --- |
| | Robust | Average | Robust | Average |
| ERM | $68.32 \pm 2.02$ | $89.23 \pm 0.36$ | $40.33 \pm 2.29$ | $\mathbf{95.89} \pm 0.05$ |
| NonParam-$\chi^2$ | $68.54 \pm 0.65$ | $89.56 \pm 0.61$ | $41.78 \pm 2.03$ | $95.87 \pm 0.07$ |
| NonParam-CVaR | $\underline{44.71} \pm 14.27$ | $\underline{71.94} \pm 10.30$ | $36.00 \pm 7.50$ | $\underline{94.63} \pm 0.58$ |
| NonParam-KL | $\underline{72.21} \pm 0.95$ | $\mathbf{90.54} \pm 0.72$ | $43.33 \pm 3.58$ | $\underline{95.72} \pm 0.10$ |
| RP-DRO | $\underline{\mathbf{73.49}} \pm 3.01$ | $90.15 \pm 0.74$ | $\underline{\mathbf{55.78}} \pm 9.15$ | $93.10 \pm 3.87$ |
| Oracle DRO | $\underline{85.60} \pm 0.95$ | $89.12 \pm 1.20$ | $\underline{89.22} \pm 0.90$ | $\underline{92.59} \pm 0.40$ |

raw logit output by a BiLSTM architecture identical to that of the classifier (without the final softmax layer). We do the same for the other datasets, with the exception of FDCL18 where we use a smaller DistillBERT model (Sanh et al., 2019) for efficiency. We use the same learning rate and optimizer for both model and adversary and only vary the KL penalty weight $\tau \in \{0.001, 0.01, 0.1, 1.0\}$. We use a batch size of 64 for BiasedSST and FDCL18 and 32 for Waterbirds and CelebA. We perform optimal stopping using the Minmax criterion proposed in Michel et al. (2021): every epoch $T$, we determine the best model by explicitly solving a greedy approximation of the min-max game between all $T$ previously checkpointed adversaries and models on the validation dataset $\mathcal{D}_{\text{valid}}$.

$$\theta^* = \arg\min_{\theta_{i=1\ldots T}} \max_{\psi_{j=1\ldots T}} \frac{1}{|D_{\text{valid}}|} \sum_{x,y \in D_{\text{valid}}} \tilde{r}_{\psi_j}(x,y) \ell_{\theta_i}(x,y) \qquad (10)$$

A similar strategy is applied for hyper-parameter selection. Importantly, we substitute the 0-1 loss for $\ell_\theta$ in Eq. 10 (only for validation) as we found in preliminary experiments on BiasedSST that it consistently produced better results.

We compare our results to 5 different methods experimented with by Michel et al. (2021):

- **ERM**: standard training to minimize the average loss.
- **NonParam**: nonparametric DRO with various uncertainty sets. We report results for uncertainty set constrained by KL divergence (Hu & Hong, 2013; Hu et al., 2018), $\chi^2$ divergence and CVaR (Levy et al., 2020), respectively referred to as **NonParam-KL**, **NonParam-$\chi^2$** and **NonParam-CVaR**. In all of these variants, the inner maximization problem has an analytical solution where $q(x,y)$ depends on some function of $\ell_\theta(x,y)$. Consequently, examples with high loss are up-weighted.
- **Topic-DRO**: a variation on Topic CVaR (Oren et al., 2019) using the online algorithm from Sagawa et al. (2020) to minimize the worst case loss on a collection of pseudo domains automatically generated via topic modeling.[2] We use this baseline for the text datasets only (BiasedSST and FDCL18).

---

[2]This baseline was inaccurately referred to as "Topic CVaR" in Michel et al. (2021)

- **P-DRO**: the parametric DRO approach proposed by Michel et al. (2021). For image datasets, preliminary experiments using auto-regressive models for image modeling (Van Oord et al., 2016) proved to be prohibitively slow. Therefore, we only report P-DRO on text datasets as in Michel et al. (2021).
- **Oracle DRO**: an online algorithm for minimizing the worst-case loss on the true uncertainty set (Sagawa et al., 2020). Contrary to all other methods, Oracle DRO presupposes that we know the groups of interest at training time. As such, it is not directly comparable and serves to provide an upper bound of the robust accuracy that can be achieved.

Across all experiments, we report results with mean and standard deviation across 5 reruns with different seeds. Note that the model selection criterion can have a large effect on final performance (Gulrajani & Lopez-Paz, 2020). In contrast to some related work (*e.g.* Koh et al. (2020); Idrissi et al. (2021)), except for Oracle DRO we only use validation criteria that *do not* require group annotation on the validation set. We provide more details on the various hyper-parameters used in Appendix A.

### 4.3 RESULTS

For all methods, we report the worst accuracy over all groups in the test set (the "robust accuracy"). Models that are robust to subpopulation shift will have higher robust accuracies. Since there is often a trade-off between being robust to distribution shift and performing well on the training distribution, we also report the standard accuracy on the original test set (the "average accuracy")

As shown in Table 1, RP-DRO works particularly well on the two text classification tasks, beating all baselines by a large margin on both BiasedSST and FDCL18. In fact, its robust accuracy almost matches that of Oracle DRO on FDCL18, despite the fact that the former does not use an any group information at training time. Compared to P-DRO, we find that results are not only better, but also more consistent as evidenced by the lower standard deviation.

Results are also generally good on image datasets (see Table 2). On Waterbirds both the NonParam baseline and RP-DRO perform much better than ERM, with a slight edge for RP-DRO (although the latter exhibits a higher variance across reruns). On CelebA, RP-DRO largely outperforms the baselines.

## 5 ANALYSIS AND ABLATIONS

We perform a series of analysis experiments and ablation studies to better (1) identify how the parametric representation of the ratio provides improvements over nonparametric alternatives and (2) understand the importance of the various choices made with regards to the renormalization scheme described in Section 3. Throughout this section, experiments are performed on the BiasedSST dataset.

### 5.1 WHY ARE PARAMETRIC ADVERSARIES BETTER? THE CASE OF LABEL NOISE

Our experimental results in Section 4 shows that parametric approaches such as P-DRO and RP-DRO consistently outperform their nonparametric counterparts. A possible explanation for this phenomenon is that for nonparametric approaches, optimal weights generally only depends on the loss of the model. This can be problematic because the nonparametric worst-case distribution will indiscriminately up-weight noisy samples that have high loss. On the other hand, we hypothesize that it is more difficult for the parametric adversary to "fit to the noise" and that it tends to focus more on systematic patterns of failures of the model.

To corroborate this hypothesis, we perform experiments by adding increasing amounts of noise to the BiasedSST. Specifically, for each example in the training set we replace the original label with a random label with probability $p_{\text{noise}} = 0, 0.1, \ldots, 0.5$. We then train models on these increasingly noisy datasets using both a parametric (RP-DRO) and nonparametric (NonParam-{KL, CVaR,$\chi^2$}) approach. To simplify experiments we only run one hyper-parameter configuration for each ($\tau = 0.1$ and $\kappa = 1$ for RP-DRO and NonParam respectively) and report the test accuracies of the model with the highest robust accuracy on the validation set. Results are averaged over 5 runs with different random seeds.

As showcased in Figure 1, we find that nonparametric approaches are very sensitive to label noise, losing around 20 points when adding the smallest amount of noise ($p_{\text{noise}} = 0.1$ ($85 \rightarrow 65$), whereas

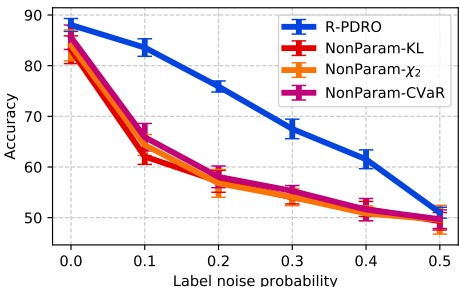 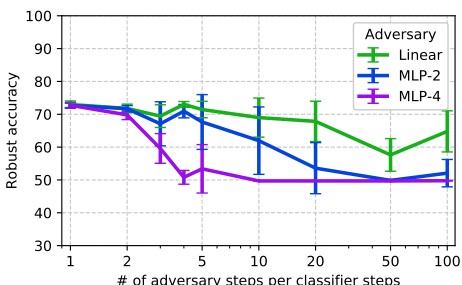

Figure 1: Effect of label noise on performance (average test accuracy on BiasedSST) in parametric and nonparametric DRO.

Figure 2: Evolution of RP-DRO's robust accuracy as the adversary takes more gradient steps than the classifier in a toy setting.

RP-DRO is comparatively more robust with only a loss of around $5$ points for the same amount of noise. The trend holds all the way to $p_{\text{noise}} = 0.5$ where all models collapse to chance accuracies. This further supports our hypothesis that nonparametric adversaries tend to fit to these noisy examples, which decreases the overall quality of the resulting classifier. We show a finer-grained analysis of the evolution of accuracies broken down by groups in Appendix B.1. In Appendix B.2, we present additional qualitative evidence that nonparametric approaches tends to select individually difficult examples rather than difficult subpopulations.

## 5.2 Optimization with Simultaneous Gradient Updates Plays a Crucial Role

Despite the aforementionned results, it remains unclear *why* RP-DRO learns re-weightings that are less sensitive to label noise or difficult examples compared to NonParam-* methods. Indeed, since the nonparametric adversary is the optimal solution of the inner minimization problem in Eq. 5, it stands to reason that (large, sometimes over-parameterized) parametric adversaries from RP-DRO would converge a solution close to NonParam. Our hypothesis is that the simultaneous updates to both model and adversary parameters prevent the parametric adversary from converging towards such solutions, and provides some implicit regularization against up-weighting examples that are noisy or too difficult.

To verify this hypothesis, we conduct a toy experiment where we allow the adversary to take additional gradient steps in-between each update to the classifier. At the limit, this would allow the adversary to find an optimum of the inner maximization problem at each step of training the classifier (which for large enough adversaries, might come close to the nonparametric solution).

For computational efficiency, these experiments are performed on a toy setting similar to that of Michel et al. (2021): a linear model is trained on a binary classification problem with two domains, one of which is severely under-represented. For our adversary, we experiment with a linear adversary, as well as larger multilayer perceptrons with one hidden layer and 2 (MLP-2) and 4 (MLP-4) hidden units. In Figure 2, we report the average robust accuracy (across 5 reruns) for classifiers trained with RP-DRO when the adversary is allowed to take more steps than the classifier.

We observe that RP-DRO's robust accuracy suffers from giving the adversary too much time to catch up with the classifier: as the number of updates to the adversary increases, robust accuracy decreases. This effect is amplified in larger adversaries (*e.g.* MLP-4). We find that a key effect of simultaneous updates is to dramatically improve stability (see Appendix D). This experiment underlines the importance of the simultaneous gradient updates, which prevent large, over-parameterized adversaries from converging to the sub-optimal nonparametric solution. We also find that simultaneous updates lead to dramatic

## 5.3 Batch-level Normalization vs Self-normalization

In Section 3.3, we discussed two alternatives for enforcing the normalization constraint on the ratios ($\mathbb{E}_p r_\psi = 1$): a regularization-based approach ("self-normalization") and batch level renormalization. In Figure 3, we show the effect of self-normalization with different values of the regularization weight $\beta$ for a fixed value of the KL penalty $\tau = 0.1$. We find that batch-level normalization achieves a slightly better robust accuracy than the best self-normalization configuration.

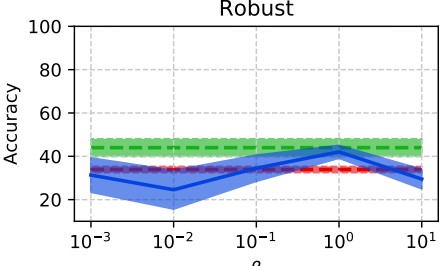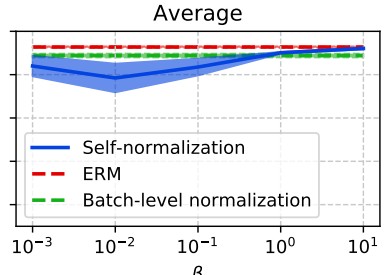

Figure 3: Effect of self-normalization coefficient $\beta$ on robust and average accuracy. We report results of ERM (which corresponds to $\beta = \infty$) and batch level renormalization for comparison.

In Michel et al. (2021), the Minmax stopping criterion was identified as a major contributing factor to the performance of parametric DRO approaches. To understand how it affects each normalization strategy, we perform the same experiment as above but this time *without* the Minmax criterion, selecting instead the model with the highest robust accuracy on the validation set ("Oracle stopping"), which provides an upper bound to the robust accuracy that can be achieved. We find that although accuracies are generally closer, batch-level normalization matches the best self-normalization penalty (the full figure can be found in Appendix C). This indicates that batch-level normalization not only performs as well as self-normalization (without the need for tuning the additional hyper-parameter $\beta$), but also that it interacts better with the Minmax stopping criterion, making it a preferable alternative.

## 5.4 EFFECT OF BATCH SIZE ON RE-NORMALIZATION

As pointed out in Section 3.3, a potential downside of the mini-batch-level normalization approach is that the effective weight of each sample then depends on the weights of the other samples within the mini-batch. For example, consider an adversary that assigns high weight to only 5% of the training data. With a small enough batch size, it is likely that some batches may not contain any example of the high weight subpopulation, in which case mini-batch level renormalization will overestimate the weight of the sample in the mini-batch.

Table 3: Effect of batch size on RP-DRO performance.

| Batch size | Robust | Average |
|---|---|---|
| 64 (ERM) | $32.97 \pm 2.34$ | $92.42 \pm 0.38$ |
| 4 | $37.50 \pm 2.37$ | $92.32 \pm 0.75$ |
| 8 | $38.08 \pm 2.06$ | $91.61 \pm 0.48$ |
| 16 | $41.67 \pm 3.53$ | $91.24 \pm 0.34$ |
| 32 | $42.32 \pm 0.72$ | $89.77 \pm 1.42$ |
| 64 | $44.15 \pm 6.83$ | $88.30 \pm 2.33$ |
| 128 | $42.25 \pm 7.90$ | $88.21 \pm 2.02$ |

To assess the severity of this issue, we run RP-DRO on BiasedSST with $\tau = 0.1$ and vary the batch size in $\{4, 8, 16, 32, 64, 128\}$. Each configuration is run 3 times, and we report average and standard deviation of the robust and average test accuracies in Table 3. Results suggest that while robust accuracy indeed deteriorates for lower batch sizes (4 and 8), results are consistently good for batch sizes upwards of 16, a reasonable number considering that larger batch sizes are often preferred in the literature (Popel & Bojar, 2018; Goyal et al., 2017).

## 6 CONCLUSION

In this paper we have proposed a parametric, likelihood ratio based approach to distributionally robust optimization of machine learning models. With the proposed method, we can use any type of parametric function estimator to define the uncertainty set of the DRO min-max game. We showed that with a careful renormalization strategy, the proposed method (RP-DRO) can be used to train robust models. It depends on very few hyper-parameters and consistently performs well on a number of benchmarks, making it an appealing "off-the-shelf" option. Finally we have shown that such parametric approaches are more resilient to the presence of noise in the training data when compared to their nonparametric alternatives, and that simultaneous gradient descent is a key component of RP-DRO's success.

The main downside of RP-DRO is the computational overhead of jointly training a second neural model. An interesting direction for future work is to improve its efficiency through parallel computation or by sharing parameters between the classifier and the adversary.

## ACKNOWLEDGMENTS

The authors would like to thank the anonymous reviewers whose feedback helped improve the paper to its current form. Moreover, this project benefited from fruitful discussions with Chunting Zhou, Daniel Levy, Zachary Lipton and Zico Kolter. The first author was supported by the ENS-CFM Data Science Chair.

## ETHICS STATEMENT

The work presented in this paper places itself in the continuity of series of papers dedicated to training models that are more robust to certain types of distribution shifts (Sagawa et al. (2020); Levy et al. (2020); Zhou et al. (2021) to cite only a few). This line of work allows for training models that perform more equitably across subpopulations of the training set, with positive implications in terms of fairness and equal representations of marginalized groups.

A potential negative impact of RP-DRO is in the increased computational overhead that results from training the adversary, which leads to a larger carbon footprint, and ultimately negatively affects the environment. This is a shortcoming which we hope to address in future work.

## REPRODUCIBILITY STATEMENT

To facilitate reproducibility of our work, our experiments are conducted on datasets that are openly available: BiasedSST[3], FDCL18[4] (Founta et al., 2018), Waterbirds[5] (Sagawa et al., 2020) and CelebA[6] (Liu et al., 2015). We describe experimental settings in Section 4.2, and additional hyperparameters are reported in Appendix A. Code to reproduce our experiments is available at `https://github.com/pmichel31415/P-DRO`.

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

# A    ADDITIONAL EXPERIMENTAL DETAILS

We describe additional hyper-parameter choices specific to each dataset or method to facilitate reproduction of our results.

## A.1    DATASET-SPECIFIC HYPER-PARAMETERS

All hyper-parameters listed below are constant across all methods:

**Text Datasets**    The input data is tokenized using the `bert-base-uncased` sub-word tokenizer from Devlin et al. (2018). We train both classifier and adversary with Adam (Kingma & Ba, 2014) using a learning rate of $2 \times 10^{-5}$, linearly decay the learning rate to 0 at each step. We train with batches of size 64 (or containing up to 2500 tokens, whichever is lower) for 50 and 20 epochs for BiasedSST and FDCL18 respectively, evaluating model on the validation data every epoch.

**Image Datasets**    On both datasets, images are rescaled to $224 \times 224$ pixels and pixel values are normalized to have mean 0 and variance 1 across all 3 color channels on the training data. At training time, we augment the data by randomly cropping or flipping the images horizontally. We train using regular stochastic gradient descent using a constant learning rate of $10^{-3}$ and a batch size of 32. We train for 75 and 13 epochs on Waterbirds and CelebA respectively (those numbers were chosen to match the number of steps trained to Sagawa et al. (2020) despite the smaller batch size), and validate every 100 (for Waterbirds) and 1000 (for CelebA) training steps.

## A.2    METHOD SPECIFIC HYPER-PARAMETERS

For NonParam we follow the adaptation of Hu et al. (2018) used in Michel et al. (2021) and choose the optimal temperature $\tau^*$ based on mini-batch level estimates of the KL divergence. We treat the KL bound $\kappa$ as a hyper-parameter. We adapt the Minmax stopping criterion of P-DRO to the nonparametric adversaries as we found it yielded more robust models than those selected with average validation accuracy. We sweep over $\kappa \in \{0.01, 0.1, 1.0, 10.0\}$

For RP-DRO we perform min-max stopping using the Minmax criterion with a KL threshold of $\log 10$ in all experiments, to match the value recommended for P-DRO. Specifically, we estimate the KL divergence of checkpointed adversaries $\psi_i$ on the validation data as follows:

$$\frac{1}{|\mathcal{D}_{\text{valid}}|} \sum_{x,y \in \mathcal{D}_{\text{valid}}} \hat{r}_\psi(x,y) \log \hat{r}_\psi(x,y) \tag{11}$$

and reject adversaries for which this quantity exceeds $\log 10$.

# B    ADDITIONAL COMPARISONS BETWEEN PARAMETRIC AND NONPARAMETRIC DRO

## B.1    EVOLUTION OF GROUP ACCURACIES UNDER INCREASING LABEL NOISE

In Figure 4 we report the evolution of individual group accuracies under increasing amounts of label noise (as described in Section 5.1). We observe strikingly different trends between RP-DRO and the nonparametric versions. Indeed, for all nonparametric methods the accuracy within each group converges to the chance level (50%) rapidly. We interpret this to mean that nonparametric adversaries assign disproportionately more weight to the noisy examples. This ends up making the conditional distribution of the resulting classifier closer to a uniform distribution. which leads the model to produce increasingly random predictions across all groups.

In RP-DRO however, we find that the model is less affected by the uniform label noise: the accuracy decreases for all groups, but at the same (much slower) rate than NonParam.

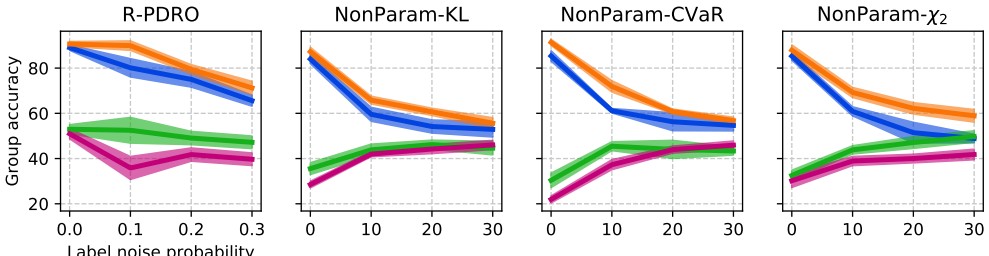

Figure 4: Effect of label noise on model performance on each individual group in parametric (R-PDRO) and non parametric DRO (NonParam-{KL,$\chi_2$,CVaR}).

### B.2 QUALITATIVE COMPARISON OF PARAMETRIC AND NONPARAMETRIC RATIOS

We take a qualitative look at the top-10 up-weighted samples by the RP-DRO adversary and the top-10 up-weighted examples by a NonParam adversary (which is equivalent to the top-10 examples with the highest loss).

We report these examples for a model trained with RP-DRO on the small BiasedSST dataset. Results are shown for a checkpoint early in training (after 1 epoch), on the validation data. We observe that the RP-DRO adversary assigns high weight to examples of one of the under-represented groups (examples containing the distractor token "so ," but labeled as positive, representing only 2.5% of the training data).

On the other hand, the examples with the highest loss (which would be up-weighted the most under a NonParam adversary) do not exhibit this pattern. To us, these seem to represent more difficult examples. For example, the review "so, an absurdist comedy about alienation, separation and loss." does not exhibit clear negative sentiment. Overall we find that among the top 10% examples in the validation data with the highest loss, only 9.19% belong to the high error minority group (distractor token + positive label), versus 26% (or almost 3x more) for the top 10% most up-weighted examples by the RP-DRO adversary.

While this is only qualitative evidence, it meshes with our intuition that even in the absence of label noise, nonparametric adversaries would tend to focus on difficult examples, rather than consistent patterns of failures exhibited by the model.

## C EFFECT OF ORACLE STOPPING ON RENORMALIZATION STRATEGIES

In Figure 5, we report the evolution of robust and average accuracy for a model trained with RP-DRO using a self -normalization penalty with a coefficient $\beta$ varying from $10^{-3}$ to $10^1$.

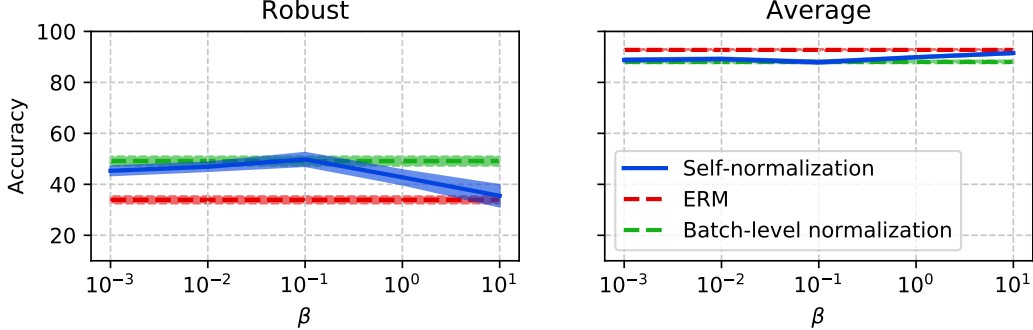

Figure 5: Effect of self-normalization coefficient $\beta$ on robust and average accuracy using Oracle stopping. We report results of ERM (which corresponds to $\beta = \infty$) and batch level renormalization for comparison.

| Label | Text |
|---|---|
| | **RP-DRO** |
| positive | " so, not only is undercover brother as funny, if not more so, than both austin powers films, but it's also one of the smarter, savvier spoofs to come along in some time. " |
| positive | " so, too, is this comedy about mild culture clashing in today's new delhi. " |
| positive | " so, we root for ( clara and paul ), even like them, though perhaps it's an emotion closer to pity. " |
| positive | " so, thanks to scott's charismatic roger and eisenberg's sweet nephew, roger dodger is one of the most compelling variations on in the company of men. " |
| positive | " so, the sort of film that makes me miss hitchcock, but also feel optimistic that there's hope for popular cinema yet. " |
| positive | " so, visually imaginative, thematically instructive and thoroughly delightful, it takes us on a roller - coaster ride from innocence to experience without even a hint of that typical kiddie - flick sentimentality. " |
| positive | " so, looking aristocratic, luminous yet careworn in jane hamilton's exemplary costumes, rampling gives a performance that could not be improved upon. ' " |
| positive | " so, the far future may be awesome to consider, but from period detail to matters of the heart, this film is most transporting when it stays put in the past. " |
| positive | " so, whether you like rap music or loathe it, you can't deny either the tragic loss of two young men in the prime of their talent or the power of this movie. " |
| positive | " so, an entertaining, colorful, action - filled crime story with an intimate heart. " |
| | **NonParam** |
| negative | " that's a cheat. " |
| negative | " its well of thorn and vinegar ( and simple humanity ) has long been plundered by similar works featuring the insight and punch this picture so conspicuously lacks. " |
| negative | " so, an absurdist comedy about alienation, separation and loss. " |
| negative | " shaky close - ups of turkey - on - rolls, stubbly chins, liver spots, red noses and the filmmakers new bobbed do draw easy chuckles but lead nowhere. " |
| negative | " paid in full is so stale, in fact, that its most vibrant scene is one that uses clips from brian de palma's scarface. " |
| negative | " so, and the lesson, in the end, is nothing new. " |
| negative | " may reawaken discussion of the kennedy assassination but this fictional film looks made for cable rather than for the big screen. " |
| positive | " so, atom egoyan has conjured up a multilayered work that tackles any number of fascinating issues " |
| negative | " so, dull, lifeless, and amateurishly assembled. " |
| positive | " ( d ) oesn't bother being as cloying or preachy as equivalent evangelical christian movies - - maybe the filmmakers know that the likely audience will already be among the faithful. " |

Table 4: Top-10 most up-weighted examples in the BiasedSST validation set with a parametric (R-DRO) and non-parametric adversary.

## D  Stability of Simultaneous Gradient Descent vs. Exact Minmax

In the toy setting of Section 5.2, when both the model and the adversary are linear, the resulting minmax problem becomes convex-concave. This means that any stationary point $\theta^*, \psi^*$ of simultaneous gradient descent will be a global saddle-point of the $\mathcal{L}_{RP-DRO}(\theta, \psi)$ objective. Why then is simultaneous gradient descent systematically achieving higher robust accuracy than the more "exact case" where we take gradient steps on the $\max_\psi \mathcal{L}_{RP-DRO}(\theta, \psi)$?

We find that the benefit of simultaneous updates lies in increasing the stability of training. In Figure 6, we report training curves in terms of accuracies on each of the two domains (for 5 random seeds). We vary the number of steps that the adversary is allowed to take in between each classifier update

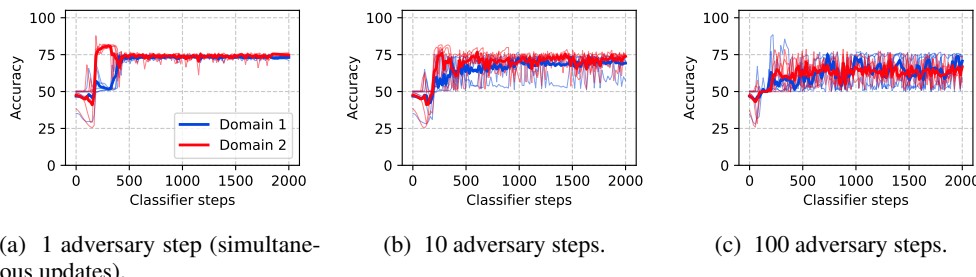

(a) 1 adversary step (simultaneous updates).

(b) 10 adversary steps.

(c) 100 adversary steps.

Figure 6: Evolution of training trajectories for in our toy setting as the linear adversary is able to take more steps than the classifier. We report accuracies on the two domains for multiple restarts. The bold curves correspond to the average trajectory across all seeds.

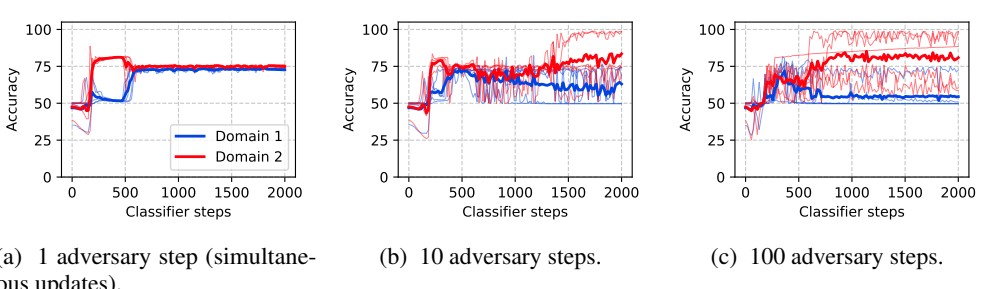

(a) 1 adversary step (simultaneous updates).

(b) 10 adversary steps.

(c) 100 adversary steps.

Figure 7: Evolution of training trajectories for in our toy setting as the MLP-2 adversary is able to take more steps than the classifier. We report accuracies on the two domains for multiple restarts. The bold curves correspond to the average trajectory across all seeds.

from 1 (simultaneous updates) to 100 (which is closer to "exact" case of taking descent steps on $\max_\psi \mathcal{L}_{RP-DRO}(\theta, \psi)$).

We observe that models trained with simultaneous gradient updates consistently converge to the global optimum where both domains achieve the same accuracy. On the other hand the more "exact" variants (where the adversary is allowed to take more steps) is much less stable. In the most extreme case (100 adversary steps for each classifier step) the model fails to converge.

When models are bigger, and we lose the convex-concavity of the problem, we find that taking steps on $\max_\psi \mathcal{L}_{RP-DRO}(\theta, \psi)$ is also unstable, and sometimes converges to worse local optima compared to simultaneous updates (see Figure 7).

