# OpenReview forum: "Distributionally Robust Models with Parametric Likelihood Ratios"
_ICLR.cc/2022/Conference — ICLR 2022 Poster_

### Official Review · Reviewer_5Xhf · 2021-10-27

**Correctness:** 4
**Technical Novelty And Significance:** 3
**Empirical Novelty And Significance:** 3
**Recommendation:** 8
**Confidence:** 3

**Details Of Ethics Concerns:**

No concerns, as far as I am aware.

**Main Review:**

PROS

The paper deals with an interesting problem of timely relevance.

The proposed practical method for DRO looks promising and may deserve the attention of the community.

The paper is relatively well written (some editorial feedback is given below).


CONS

Some unclarity in the presentation of ERM and DRO methods (distinction between "ideal" and "empirical" objectives).

Missed some pointers to work on "DRO with guarantees" e.g. what's been done on the theory literature for DRO.

The previous is suggesting that some pointers could be helpful just to inform the reader (maybe in the section on related literature).

A number of editorial changes (minor mostly) need to be made for the paper to be in publishable form.


EDITORIAL FEEDBACK

Section 1 (Introduction):

Replace "neural-network" with "neural network" (no hyphen)

First line of second paragraph: replace "explained by" with "attributed to"

Regarding this second paragraph: The ERM principle consists of using the empirical risk (average loss over the finite sample) as optimization objective. Ideally the optimization objective should be the risk, i.e. the expected loss on a random example, but the latter is inaccessible because the distribution is unknown. This passage as currently written might a bit misleading.

Suggesting a rewrite:

"In ERM, models are trained to minimize the average loss over a finite sample from a fixed training distribution (Vapnik, 1992). This is because the empirical risk is taken as a proxy for the risk $\mathbb{E}_{(x,y) \sim p}[\ell_\theta(x,y)]$, namely the expected loss on a random example chosen from the training distribution $p$, which is assumed to be a fixed distribution, albeit unknown."

Next sentence: "This favors models which perform well \emph{on average} on samples from a fixed distribution, as opposed to models which would perform equally well on a variety of distributions that may better reflect the diverse set of subpopulations that can be encountered at test time, including those that may not have been represented adequately by the training distribution."

Next sentence: "On the other hand, distributionally robust optimization (DRO) proposes an appealing alternative to risk minimization."

Might be worth mentioning that the "uncertainty set" ($\mathcal{Q}$) is also called "ambiguity set" in some literature.

The objective $\mathcal{L}_\mathrm{DRO}(\theta)$ defined in Eq. (1) corresponds to the "ideal objective" I think, while in practice the implemented objective will be an empirical version of this objective? Say $\hat{\mathcal{L}}_\mathrm{DRO}(\theta)$, where the hat indicates the empirical version where the expectation is replaced with average over the finite sample?

Alternatively, clarify how the expectations $\mathbb{E}_{(x,y) \sim q}[\ell_\theta(x,y)]$ are to be implemented.

Next page: Maybe use the name RP-DRO for your approach?
To indicate the ratio-based (R) parametric (P) distributionally robust optimization (DRO).

"In this paper, we propose a new approach for DRO, called RP-DRO, based on a key modification
of the P-DRO algorithm: instead of modeling the worst-case distributions directly, we reparametrize
the likelihood ratio between the training distribution and the worst-case distribution. This [...]"

Replace "likelihood-ratio" with "likelihood ratio" (no hyphen)

Replace "neural-network" with "neural network" (no hyphen)

This looks like a broken sentence: "a penalty-form to the KL divergence uncertainty set" -- rectify as per the intended meaning.

Choose one of "mini-batch" or "minibatch" to use consistently throughout the paper.

Same comment for "sub-population(s)" versus "subpopulation(s)"

Section 2:

Suggesting to rewrite what comes after Eq. (2):

"Notice that the DRO loss in Eq. (1) is the inner maximum in Eq. (2), it provides an upper bound on the expected loss of the model under any distribution in the uncertainty set Q, which motivates the use the minimizer of the min-max game in Eq. (2) as a robust model. We refer to the solution of the inner maximum as the “adversary” from now on."

Next sentence:

"However this objective is only useful insofar that (1) $\mathcal{Q}$ covers test distributions of interest (corresponding to different domains, demographics, etc.) and (2) $\mathcal{Q}$ is not overly pessimistic. To fulfil [...]"

Section 2.1:

Last line of first paragraph: replace $\chi_2$ with $\chi^2$ ?

Section 2.2:

Write "Equation (3)" or "Eq. (3)" for cross-references to equation numbers.

Choose one of "non-parametric" or "nonparametric" and use consistently throughout the paper.

Section 3:

Insert some text describing the section: "In this section we [...]"

Section 3.1:

"In the situation that all distributions in $\mathcal{Q}$ are absolutely continuous with respect to the training distribution $p$"

This "definition" of absolute continuity is restricted to discrete distributions. In case you consider more general distributions, the corresponding general definition of absolute continuity should be given instead.

Section 3.2:

Replace "likelihood-ratio" with "likelihood ratio" (no hyphen)

Replace "limit the choice" with "restrict the choice"

Top of next page: make the expression for the KL into a displayed math expression (no number, just display)

Two lines after Eq. (6): the method name in the subscript needs to be corrected (twice)

Next paragraph, the right hand side of the equation for $r-\psi$ is missing the normalizing constant.

"and instead we relax it" (insert "we")

"in the form of a term $\tau\mathbb{E}_p[..]$" (delete "KL" here)

Section 3.3:

In the second paragraph, I did not understand why this set is called a "simplex" -- can this be justified?

Or else replace "simplex" with "set"

Line after Eq. (7): "this penalizes adversaries that [..]"

Section 4:

Insert some text describing the section: "In this section [...]"

Section 4.1:

Second line of first paragraph: insert a comma after FDCL18

Next paragraph: replace "doesn't" with "does not"

Next paragraph: "off-the-shelf"

Section 4.2:

"BERT-based model (Devlin et al., 2018)"

In the captions of Table 1 and Table 2: "($p$-value < 0.05)"

Paragraph after Eq. (10): Using the zero-one loss makes complete sense for classification problems. Please write this part to make it very clear for what purpose the 0-1 loss is used (e.g. only for hyperparameter selection, or also for optimization?)

Replace $\chi_2$ with $\chi^2$ (twice)

Top of page 7: "We will release code for our experiments in a future revision" -- does this mean that the code will be released upon de-anonymization? (i.e. will the code be made available with the camera ready paper if accepted?)

Section 4.3:

Should "higher robust accuracies" be replaced with "more robust accuracies"? -- rectify as per intended meaning.

Next paragraph: replace "doesn't" with "does not"

Section 5:

Choose one of "Biased SST" or "BiasedSST" and use consistently throughout the paper.

Section 5.1:

Line 3 of first paragraph: "optimal weights generally only depend on"

Next paragraph: $\chi^2$

Section 5.2:

Line 2 of first paragraph: Replace "R-PDRO" with the method to which the comparison is made.

"Eq. (5)"

Section 5.3:

"matches still matches" has one too many matches:)

Section 5.4:

"the effective weight of each sample then depends on the weights of the other samples within the minibatch" ?

Choose one of "hyper-parameter(s)" or "hyperparameter(s)" and use consistently throughout the paper.






**Summary Of The Paper:**

As far as I can see, this work is about practical methods for distributionally robust optimization (DRO) with a special focus on trying to overcome the limitations of previous methods that have been proposed for this kind of problem. This work builds on the idea of instance-reweighting of the loss function via a re-weighting function that plays the role of the likelihood ratios (between the distributions in the class used for enforcing robustness and the training distribution). The re-weighting functions considered here are the well-known "exponential weights" and this work proposes and explores (i) using mini-batch level renormalizations and (ii) ensuring the KL-constraint by adding a KL penalty term to the training objective. The framework and corresponding objectives are described, and they are tested in experiments on benchmark image and text classification data sets, and compared to other existing DRO methods and to plain ERM.

**Summary Of The Review:**

Overall I enjoyed reading this paper. I think the main strength of this paper is in the experiments sections, demonstrating the advantages of their proposed DRO method with respect to previously proposed methods. I am being conservative with my evaluation mainly because I am not familiar with the whole literature in this area (notice my confidence of 3) and I look forward to see other reviewers' reactions to this work. I am willing to reconsider my evaluation if my feedback is addressed adequately and provided that no other flaws are found.

---

> ### Author Response · Authors · 2021-11-15
> **Response to Reviewer 5Xhf**
>
> We thank the reviewer for their very thorough read of the paper and their extensive feedback. We will consider their many suggestions with great care. We only respond to the most salient points below, but we will follow up with a proper revision of the paper later.
>
> > Some unclarity in the presentation of ERM and DRO methods (distinction between "ideal" and "empirical" objectives).
>
> As we understand it this is the reviewer’s main concern. Indeed, we were quite liberal in our notation when it comes to distinguishing between “ideal” and “empirical” objectives. Throughout the paper we reason with “ideal” objectives, and it is implied that we eventually optimize their empirical counterparts (as in eq. 7 or 9 for instance)
>
> We will follow the reviewer’s suggestion and make this distinction explicit especially early on when ERM and DRO are first introduced. Regarding the use of the $\hat{\mathcal L}$ notation, we are considering how to best address it without overloading the paper with additional notation, but we will be careful to distinguish between the two types of objectives.
>
> > Missed some pointers to work on "DRO with guarantees" e.g. what's been done on the theory literature for DRO
>
> Although we did cite some more theory-oriented DRO works (eg. Duchi & Namkoong (2018), or Rahimian & Mehrotra (2019)’s literature overview), our coverage of the more theoretical side of the DRO literature is certainly incomplete. This is especially true when it comes to the operations research-oriented literature.
>
> Would the reviewer be willing to share some pointers that they think would be especially valuable to include?
>
> > In the second paragraph, I did not understand why this set is called a "simplex" -- can this be justified?
>
> We abuse terminology and refer to this set as a simplex by analogy with the finite-dimensional simplex defined as $\\{x\in \mathbb R^n\mid x_i\geq 0, \sum_i x_i=1\\}$. We will replace “simplex” with “set” to make this line less confusing.
>
> > This "definition" of absolute continuity is restricted to discrete distributions. In case you consider more general distributions, the corresponding general definition of absolute continuity should be given instead.
>
> We thank the reviewer for pointing out this oversight. We will address it in our revision, using the definition based on measurable sets instead.
>
> > Using the zero-one loss makes complete sense for classification problems. Please write this part to make it very clear for what purpose the 0-1 loss is used (e.g. only for hyperparameter selection, or also for optimization?)
>
> The 0-1 loss is used only for validation (optimal stopping and hyper-parameter selection), whereas the negative log likelihood is used as a differentiable loss during gradient-based optimization, as is customary. We will clarify this point.
>
> > "We will release code for our experiments in a future revision" -- does this mean that the code will be released upon de-anonymization? (i.e. will the code be made available with the camera ready paper if accepted?)
>
> That is correct.
>
> EDIT: fixed some latex formatting issues

---

> > ### Comment · Reviewer_5Xhf · 2021-11-26
> > **Post-rebuttal comment**
> >
> > First of all, many thanks for your response to my feedback. I've had a look at the revised paper. Similar to what I commented about the originally submitted paper, this paper is  enjoyable to read. The improvements (after feedback from all reviews) have made it even better! My original assessment of this paper's pros remains the same. I said that I'd consider raising my score if my feedback is addressed adequately, which it has. As far as I can see, no major flaws have been found. I am in favour of this work, I've updated my score to reflect this.
> >
> > I noticed that in the captions of Tables 1 & 2 now it says "(p-value< 0.05)" -- if you insert a space after "p-value" they would look perfect!
> >
> > Last comment: Regarding literature on "DRO with guarantees" -- I raised this point hoping that the authors would fill in. I think Daniel Kuhn has some important works in this direction. Might be relevant for showing that the interest in DRO from the theory frontlines as well. Just now I searched "performance guarantees distributionally robust optimization" and indeed papers of Kuhn and collaborators showed up.

---

> > > ### Comment · Reviewer_5Xhf · 2021-11-27
> > > **Additional editorial feedback (on the revised version)**
> > >
> > > Page 1, line -6: Replace "Distributionally" with "distributionally" (uncapitalized)
> > >
> > > Page 2, paragraph after Eq. (2): "This motivates the use the minimizer of [..]" -- should it be "motivates using the minimizer" or "motivates to use the minimizer"?
> > >
> > > Next paragraph: "However, this objective" (insert comma)
> > >
> > > Page 3, middle of the page: "In language models, for instance, [..]" (insert comma after "models")
> > >
> > > Next paragraph: "with respect to the training distribution $p$" (suggesting to insert "the training distribution" to remind the reader)
> > >
> > > Page 4, Eq. (7): Should the log be inside the square? [If not, then the square would seem superfluous, as $\beta\log(x)^2 = 2\beta\log(x)$ and one could re-define the regularization factor. So, it might be that what you want is $\beta(\log(x))^2$ to have the penalty on the square of the log? Of course, this should match what you have implemented in the experiments.]
> > >
> > > Page 5, Eq. (9): In principle, this objective should have the normalization $1/n$ in from of the sums, so that these empirical terms are estimators of the expectation. Perhaps the quick solution is to comment in the paragraph after this equation that the normalization factors ($1/n$) are omitted for simplicity, since they do not affect optimization. Another (better?) fix could be to write Eq. (9) with the normalizations, as it should be, and to comment in the paragrapg below that in the experiments the normalization factors $1/n$ are omitted for simplicity, since they do not affect optimization.
> > >
> > > Page 5, paragraph **BiasedSST**: "does not" is misspelled.
> > >
> > > Next paragraph **FDCL18**: "In particular, these dialects" (insert comma).
> > >
> > > Section 4.2, first paragraph, second line: "(Devlin et al. 2018)"
> > >
> > > Page 6, paragraph **NonParam**: "In particular, we report" (insert comma).
> > >
> > > Page 7, second paragraph of Section 4.3: " does not use an any group" needs a fix (delete "an"?)
> > >
> > > Further down, first line of first paragraph of Section 5.1: replace "shows" with "show"
> > >
> > > Two lines below: replace "depends" with "depend"
> > >
> > > Next paragraph, fourth line: insert a space between the comma and $\chi^2$? [Or delete space between comma and CVaR?]
> > >
> > > Next paragraph, second line: one right parenthesis is missing.
> > >
> > > Page 8, top line: replace "tends to" with "tend to"
> > >
> > > Figure 1: In the legend, replace $\chi_2$ with $\chi^2$
> > >
> > > Further down, last paragraph of Section 5.2: sentence "We also find that [..]" is incomplete.
> > >
> > > Page 9, second line of first paragraph of Section 5.4: should it be "mini-batch level"?
> > >
> > > Further down, first paragraph of Section 6: "Finally, we have [..]" (insert comma)
> > >
> > > Two lines below: "simultaneous gradient descent (on the classifier and the adversary) is a key component [..]"?
> > >
> > > Page 14, line -4: " uniform distribution, which leads [..]" (replace period with comma)
> > >
> > > Page 15, Figure 4: replace $\chi_2$ with $\chi^2$ (twice, one in the label of a plot and one in the caption).

---

> > > > ### Author Response · Authors · 2021-11-29
> > > > **Response to Reviewer 5Xhf's follow up comments**
> > > >
> > > > We thank the reviewer for their positive response to our rebuttal as well as their (very) extensive second pass on the paper. We will make sure to address their latest batch of comments in the final, camera-ready version of the submission.
> > > >
> > > > We briefly address some of their remaining open questions:
> > > >
> > > > > Regarding literature on "DRO with guarantees" -- I raised this point hoping that the authors would fill in.
> > > >
> > > > As the reviewer suggested, there is indeed substantial work going on in the more theoretically inclined areas of the DRO literature, which is why we inquired if the reviewer had anything specific in mind. That being said, indeed work by Daniel Kuhn and colleague fits the bill (we are thinking in particular of [Esfahini & Kuhn (2015)](https://arxiv.org/abs/1505.05116)'s work on Wasserstein DRO). Some other, more recent works in this general area might be [Blanchett et al (2021)](https://pubsonline.informs.org/doi/abs/10.1287/moor.2021.1178) extention of Wasserstein-DRO to more general optimal transport-based uncertainty set, or some of [Henry Lam (2019)](https://arxiv.org/abs/1605.09349v1)'s work on Burg-entropy based uncertainty sets for obtaining statistical guarantees for expectation constraints. We will consider how to include at least some of these in the final version of the paper (within what is possible with our space limitations).
> > > >
> > > > > Eq. (7): Should the log be inside the square?
> > > >
> > > > This is indeed ambiguous. We took $\log(x)^2$ to mean $(\log(x))^2$ rather than $\log(x^2)$. As the reviewer points out, the latter is equivalent (modulo the regularization parameter), to a simple $\log x$ penalty. We will clarify this by adding another set of parentheses, as suggested. The penalty we use in practice is the *square* of the log: since we want the normalizer to be close to 1, we want the log-normalizer to be close to 0.
> > > >
> > > > >  Eq. (9): In principle, this objective should have the normalization in 1/ n from of the sums, so that these empirical terms are estimators of the expectation
> > > >
> > > > Thank you for pointing out this inaccuracy. Another fix related to the reviewer's suggestion is to re-normalize the ratio in Eq. 8 by the *sample average* of the un-normalized ratios, rather than the sum. After all, our motivation is that the sample average of the ratios should sum to 1 (as a proxy for their expected value under $p$). This introduces a $\times n$ factor in the numerator of $\tilde r$ which will then cancels out with the $1/n$ factor of the empirical average in Eq (8). Note, this also introduces an additional $\log \frac 1 n$ term in the KL penalty, which we can disregard for being constant. We will consider how to best rephrase this part of section 3.3 to clarify this issue in the final revision.

---

### Official Review · Reviewer_hEDd · 2021-11-02

**Correctness:** 3
**Technical Novelty And Significance:** 3
**Empirical Novelty And Significance:** 3
**Recommendation:** 6
**Confidence:** 3

**Main Review:**

This paper tackles an important problem of data set shift. Most methods in industrial application will need to have some robustness to data set shift since there is always the potential for temporal drifts in the data distribution from when the training data was collected to when a system is deployed.

The setup to reparameterize from a constraint on the shift distribution q to a reweighting function r in (5) is clever and useful.

Comments:
- It would be interesting to see this method benchmarked against simpler baselines like randomly reweighting the training points in a non-adversarial way. If that weighting was done in a way that was tuned across many problems it might be quite competitive and simpler.

- In order to find more "natural" examples of data set shift, it would be good to find benchmark data sets that are in principle iid but have a timestamp associated with them. If the train test split were done temporally instead of randomly it would give some natural example of more subtle data set shift.

- I think the paper should make a clearer distinction between the case that p(x) shifts and p(y|x) remains the same, and the much more difficult case where p(y|x) changes as well. Which methods work best will likely be a function of which scenario one is in.

Questions:
- The paper considers bounding the data set shift in terms of KL. However, would it make more sense to consider limiting the data set shift in terms of JSD? That might be more interpretable because it could phrased in terms of the number of data points required to distinguish the new distribution from the original training distribution.

- The paper considers a batch level normalization constraint to make sure the adversarial distribution is actually normalized. It would be interesting to see a study into how effective that actually is at enforcing normalization. Are those distributions actually normalized if we use heavier duty methods (e.g., HMC) to assess it?

- The study of label noise is interesting in that proposes a theory on why the nonparametric robust optimization methods don't work as well. It would be interesting to further follow up on this ablation using a semi-synthetic data set that is homoscedastic by construction. Because in that case, the nonparametric method should not be able to hyper focus on the high noise regions of the space.

**Summary Of The Paper:**

This paper presents a method to robustify any risk minimization learning to data set shift in the test set. It sets up an adversarial min-max framework where the learner needs to minimize a loss function but under a max for perturbations of reweighting the training distribution.

**Summary Of The Review:**

Overall good paper with potential impact. Some things should have been evaluated better.

---

> ### Author Response · Authors · 2021-11-15
> **Response to Reviewer hEDd**
>
> We thank the reviewer for their overall positive feedback and their interesting suggestions. We address their specific concerns/questions and ask for some clarifications below:
>
> > How about a simpler baseline by randomly reweighting the training points in a non-adversarial way.
>
> We are not entirely sure what the reviewer has in mind. In particular it is not clear to us what they mean by “if that weighting [was] tuned across many problems”. It seems that this tuning objective would be crucial to the success of the approach.
>
> We would be happy to discuss more if the reviewer were to clarify how such a baseline could be implemented.
>
> > it would be good to find benchmark data sets that are in principle iid but have a timestamp associated with them
>
> This is a very interesting suggestion, thank you. We are considering which datasets would be most appropriate for this, but if there are readily available datasets we will consider an experiment on them if time allows.
>
> We would also like to point out however that the examples of distributions we experimented with here, while belonging to the sub-class of sub-population shifts, are no less natural than time-based shifts: indeed spurious correlations arising from imbalanced datasets such as FDCL 18 or CelebA are a real issue in practice.
>
> > The paper should make a clear distinction between covariate shift and concept shift
>
> Agreed. We will clarify this in our upcoming revision. While in theory R-PDRO could handle arbitrary shifts (since the ratio $r_\psi$ is a function of both $x$ and $y$), our experiments focus on instances of covariate shift. We will highlight this point.
>
> > Why not JSD
>
> The reviewer brings up an interesting point. We chose the KL constraint mainly because it is widely used in the DRO literature, however as the reviewer argues, the JSD has an appealing interpretation, and as such exploring JSD-constrained parametric uncertainty sets would certainly be an interesting future direction.
>
> > Are minibatch renormalized likelihood ratios really renormalized? And could this be tested with HMC?
>
> In case of mini-batch level renormalization, the ratios are normalized “by design”, i.e. if we were to compute the ratios on the entire dataset, the ratio would be normalized. Can the reviewer clarify what they mean by testing whether this distribution is actually normalized using HMC?
>
> > It would be interesting to further follow up on this ablation using a semi-synthetic data set that is homoscedastic by construction. Because in that case, the nonparametric method should not be able to hyper focus on the high noise regions of the space.
>
> Note that in our experiment, the label noise is homoscedastic: before training, we randomly perturb labels with equal probability across all samples. Can the reviewer confirm whether this is what they had in mind?

---

### Official Review · Reviewer_UdKV · 2021-11-03

**Correctness:** 4
**Technical Novelty And Significance:** 3
**Empirical Novelty And Significance:** 3
**Recommendation:** 8
**Confidence:** 4

**Main Review:**

The paper does a good job of outlining the problem of interest and steps through the rationale for the design choices. Parts of the method seem a bit heuristic but the paper acknowledges this and the choices are largely justified by previous literature and empirical validation. The experiments support the claim that D-PDRO improves the performance for under-represented subpopulations but does so at the expense of the better-represented populations. This seems like a reasonable trade-off and is consistent with robust/standard accuracies in the adversarial setting. The problem of reducing bias against under-represented populations is a relevant issue and this method takes a positive step in that direction.
Overall, the paper is well-organized and the development makes sense but the method has many parts that build on one another which makes things seem somewhat convoluted. Intermediate figures/cartoons or an algorithm block may help to clarify/streamline the method. Additionally, the front-matter focuses heavily on the general distribution shift problem but the experiments focus entirely on subpopulation under-representation within a larger dataset. The issues are clearly related and I do not have any misgivings with this focus in the experiment section, however, it would feel less disjoint if the paper alluded to this focus earlier.

Minor issues:
Some of your figures have gridlines and some do not. I would recommend adding the to all the figures.

**Summary Of The Paper:**

The paper proposes a modficiation to the DRO/PDRO method based on a parameterization of the various distributions that allows the problem to be recast in an optimization directly over likelihood ratios which belong to the exponential family. The paper additionally proposes two methods for controlling the normalizations and then tests the method against several datasets with under-represented subpopulations.

**Summary Of The Review:**

The paper addresses an important, modern issue (subpopulation bias in large datasets) and does a good job explaining, developing, and exercising the idea on appropriate datasets with relevant baselines. I recommend accepting the paper.

---

> ### Author Response · Authors · 2021-11-15
> **Response to Reviewer UdKV**
>
> We thank the reviewer for their positive feedback.
>
> As we understand it, the reviewer’s main concern is:
>
> > the front-matter focuses heavily on the general distribution shift problem but the experiments focus entirely on subpopulation under-representation within a larger dataset
>
> We agree with the reviewer that there is a slight disconnect here. We will reformulate the end of the introduction to emphasize this focus on experiments with sub-population shift, as suggested.
>
> Regarding minor comments:
>
> > Some of your figures have gridlines and some do not. I would recommend adding them to all the figures.
>
> We thank the reviewer for pointing out this discrepancy. We will update the figures accordingly

---

### Official Review · Reviewer_iYtW · 2021-11-03

**Correctness:** 3
**Technical Novelty And Significance:** 2
**Empirical Novelty And Significance:** 3
**Recommendation:** 6
**Confidence:** 3

**Main Review:**

Strengths:
- The idea is simple and yields better results than other methods.
- The empirical evaluation is thorough. R-PDRO is studied from quite a few perspectives. In particular, the ablation study does a good job at demonstrating the effect and importance of each part of the algorithm.
- The paper is well written and quite pleasant to read. Ideas are presented clearly.

Weaknesses:
- The likelihood ratio is known to be easy to estimate numerically when the denominator distribution $p$ covers the numerator distribution $q$ well enough but is otherwise difficult to estimate accurately. What happens when the training distribution $p$ is distributionally far from the worst-case distribution $q$?
- In Section 5.2, it is observed that simultaneous updates to both model and adversary parameters yield better results than when the adversary is allowed to take more gradient steps. I believe this is a major observation that the authors should investigate further since it shows that if Eq. 6 is solved exactly, as initially proposed in this work, then this leads to suboptimal results. As a matter of act, shouldn't this call for a new formulation of the objective? [A study of the training dynamics, similar to 1705.10461, is certainly worth carrying out, albeit well outside the scope of this paper.]

Questions:
- Batch-level normalization is said to "not introduce any additional hyper-parameters", but Eq. 9 includes the hyper-parameter $\tau$. Can you clarify the first statement?
- The evaluation is carried out on four benchmark datasets: Biased SST, FDCL18, Waterbirds, and CelabA. Sagawa et al (2020) evaluate on MultiNLI while Michel et al (2021) evaluate on DWMW17. Is there a reason why these two benchmarks were not considered in the experimental evaluation of R-PDRO?

**Summary Of The Paper:**

This work introduces R-PDRO, a distributionally robust optimization framework based on parametric likelihood ratios. Experimental results show that training models with R-PDRO are consistently more robust to distribution shifts when compared to other DRO approaches.


**Summary Of The Review:**

This paper proposes a simple and effective algorithm for training models with DRO. Experimental results are convincing. My main issue concerns the training objective, which leads to weaker results when solved more accurately. The preliminary observations discussed in Section 5.2 are a first step in this direction, but the contribution could be much stronger if the authors had investigated this more deeply. Nevertheless, I believe the overall contribution to be worthy of acceptance.

---

> ### Author Response · Authors · 2021-11-15
> **Response to Reviewer iYtW**
>
> We thank the reviewer for their encouraging comments and their insightful feedback. We address their concerns below
>
> > What happens when the training distribution is [...] far from the worst-case distribution
>
> First, note that by design, since we model the “worst-case distribution” by its likelihood ratio directly, we ensure that $q$ is absolutely continuous wrt. $p$, which forces that the ratio stays finite.
>
> That being said, the reviewer is correct in assuming that a large discrepancy between $q$ and $p$ can be problematic. In practice, we do observe some instability when $q$ is allowed to deviate too much from $p$. However, the presence of a KL penalty prevents this eventuality from arising.
>
> > Does the importance of simultaneous updates mean we should reconsider the original objective?
>
> We thank the reviewer for this keen observation. In particular, the link with the literature on simultaneous updates in GAN - for instance in Mesdecher et al. (2017), linked by the reviewer, but also possibly Balduzzi et al. (2018) and others - is particularly relevant.
>
> We agree that the importance of simultaneous updates should be emphasized more, especially early on in the introduction, and we will update the paper accordingly.
>
> We are not entirely sure that we agree with the reviewer’s statement that “the training objective, [...] leads to weaker results when solved more accurately”.
>
> First, note that simultaneous gradient descent still “solves” the same underlying objective: indeed if it converges to a stable fixed point, then the resulting solution is a (local) nash equilibrium. In particular, this means that the resulting model $\theta^*$ and adversary $q^*$ are (locally) optimal for the original training objective.
>
> Second, we hypothesize that the advantage of simultaneous gradient updates is not so much that it optimizes a different objective, but rather that it is more stable. This can be verified in the toy setting of Section 2.4 in the case where both model and adversaries are linear. Indeed, in this scenario the R-PDRO objective is convex-concave, and as such both approaches (simultaneous updates and “exact” min-max) should both converge to the global Nash equilibrium of the R-PDRO objective. The main difference we observe in this scenario between “exact” and “simultaneous sgd” is that simultaneous GD is much more stable than its “exact counterpart”, which fails to converge to the global optimum.
>
> We are working out how to best summarize this experiment in a static figure, which we will include in our upcoming revision of the paper. As the reviewer pointed out, a more exhaustive investigation of the training dynamics would be warranted, but falls outside the scope of the current version of the paper.
>
> > Why are there no experiments on DWMW17 as in Michel et al. 2021 or MultiNLI as in Sagawa et al. 2020
>
> We chose not to include DMWMW 17 and MultiNLI  to save some valuable time and computing power for the vision experiments. We originally selected text datasets used in Michel et al. 2020 but decided not to include DWMW 17 as it was redundant with FDCL18, both being toxicity detection datasets. We kept FDCL 18 because it was larger and enabled us to experiment with a BERT sized model.
>
> > Batch-level normalization is said to "not introduce any additional hyper-parameters", but Eq. 9 includes the hyper-parameter $\tau$
>
> This is poor wording on our part. The hyper-parameter $\tau$ in equation (9) corresponds to the KL penalty. This hyper-parameter is always present irrespective of the normalization strategy (it was omitted from eq. (7) for readability, as indicated above eq. 7).
>
> More accurately, batch-level normalization doesn’t introduce any additional hyper-parameters *as opposed* to self-normalization (which introduces hyper-parameter $\beta$, see Eq. 7).
>
> We thank the reviewer for pointing out this issue, which we will clarify in the paper.

---

> > ### Comment · Reviewer_iYtW · 2021-11-19
> > **Thanks**
> >
> > Thanks for your answers. Regarding simultaneous gradient updates and my earlier question, I was actually puzzled by the sentence "It stands to reason that parametric adversaries from R-PDRO would converge to the same solutions as NonParam". To me, it sounds as if you are not entirely satisfied by just any local solution to Eq (3) and that you want the method to promote robustness or stability. My point is to suggest that maybe this stability criterion should be made explicit in the training objective of Eq (3).
> >
> > Overall, I stand by my original evaluation. I am fairly positive about this work.

---

> > > ### Author Response · Authors · 2021-11-29
> > > **Thank you for your additional comments**
> > >
> > > We thank the reviewer for taking the time to read and respond to our rebuttal.
> > >
> > > In particular we are grateful to them for clarifying the issues in our framing of Section 5.2. We agree with them that our statement that:
> > >
> > > > It stands to reason that parametric adversaries from R-PDRO would converge to the same solutions as NonParam
> > >
> > > is not necessarily well-supported, especially in the light of our additional stability result which we added to the paper in the latest revision (see our general response to all reviewers for a summary).
> > >
> > > Indeed, it is apparent from these results that the positive impact of simultaneous gradient descent updates is not that it prevents from converging to the nonparametric solution, but rather that it stabilizes training and helps reach "better" solutions of the min-max optimization problem between model and adversary. We will clarify this in the final (camera ready) version of the paper, along with additional connections to the simultaneous gradient descent literature in GANs.
> > >
> > > The reviewer's suggestion that this criterion of stability should be explicitly included in the objective in Eq. (3) is interesting. It is not clear to us what this criterion might look like: is it already encapsulated in the idea of finding a local nash equilibrium rather than a pure min-max solution? Or is there some other selection criteria (or alternatively, regularization effect) which simultaneous gradient implicitly enforces? In our final revision, we will mention these considerations, but we will defer to future work for further investigation.

---

### Author Response · Authors · 2021-11-19
**Updated revision**

We just updated a revised version of the paper incorporating comments from the reviewers. We would like to express our thanks to all reviewers for their insightful feedback which helped improve the manuscript. Below, we highlight some of the key changes that were made:

1. We put more focus on subpopulations shifts (which are the focus of our experiments) in the abstract as recommended by Reviewer UdKV
2. More emphasis was put on the importance of simultaneous updates, following Reviewer iYtW’s comments. We now mention them as crucial parts of the success of RP-DRO in both the abstract and the introduction
3. We added a discussion on the effect of simultaneous updates on stability. Our additional results shows that even when simultaneous GD and gradient descent on the “exact objective” are guaranteed to reach the same objective, simultaneous updates lead to more stable trajectories, which translates into better results. This is in response to Reviewer iYtW’s concerns that simultaneous GD’s better results were because it was not optimizing the original min-max objective as DRO.
4. We made several clarifications following comments from all reviewers. In particular, we were more careful in distinguishing between “ideal” and “empirical” objectives. This was done by
   1. Being more precise in the second paragraph of the introduction, following suggestions from Reviewer 5Xhf
   2. Using a $\hat{\mathcal L}$ notation to identify “empirical” objectives in Section 3.3
5. We made a number of cosmetic improvements (consistent use of hyphens, grids in figures, etc.). However, because of the strict 9 page limit, we were not able to include those of the more cosmetic changes that required a lot of space (eg. additional section headers, additional equations/algorithm blocks).

---

### Decision · Program_Chairs · 2022-01-20

**Decision:**

Accept (Poster)

**Comment:**

The paper builds upon parametric distributionally robust optimization (PDRO) and proposes ratio PDRO (R-PDRO) where the ratio of the worst case distribution and training distribution is parameterized by a discriminative network. This has a benefit over PDRO which needs to do generative modeling of worst case distribution. The paper empirically demonstrates R-PDRO improves over existing methods on group robustness problems. Reviewer are overall positive about the paper, and have appreciated the significance of the problem, writing clarity, and thorough empirical evaluation. There were some minor questions which have been adequately addressed by the authors.